# Visual Motor Reaction Times Predict Receptive and Expressive Language Development in Early School-Age Children

**DOI:** 10.3390/brainsci13060965

**Published:** 2023-06-19

**Authors:** Areej A. Alhamdan, Melanie J. Murphy, Sheila G. Crewther

**Affiliations:** 1Department of Psychology, Counselling and Therapy, La Trobe University, Melbourne, VIC 3086, Australia; m.murphy@latrobe.edu.au; 2Department of Psychology, Imam Mohammad Ibn Saud Islamic University, Riyadh 11564, Saudi Arabia; 3Centre for Human Psychopharmacology, Swinburne University of Technology, Melbourne, VIC 3122, Australia

**Keywords:** early school-age children, development, NVIQ, language, multisensory processing, motor reaction times, visuo-motor processing

## Abstract

Proficiency of multisensory processing and motor skill are often associated with early cognitive, social, and language development. However, little research exists regarding the relationship between multisensory motor reaction times (MRTs) to auditory, visual and audiovisual stimuli, and classical measures of receptive language and expressive vocabulary development in school-age children. Thus, this study aimed to examine the concurrent development of performance in classical tests of receptive (Peabody Picture Vocabulary Test; abbreviated as PPVT) and expressive vocabulary (Expressive Vocabulary Test; abbreviated as EVT), nonverbal intelligence (NVIQ) (determined with the aid of Raven’s Colored Progressive Matrices; abbreviated as RCPM), speed of visual–verbal processing in the Rapid Automatic Naming (RAN) test, Eye–Hand Co-ordination (EHC) in the SLURP task, and multisensory MRTs, in children (*n* = 75), aged between 5 and 10 years. Bayesian statistical analysis showed evidence for age group differences in EVT performance, while PPVT was only different for the youngest group of children aged 5–6, supporting different developmental trajectories in vocabulary acquisition. Bayesian correlations revealed evidence for associations between age, NVIQ, and vocabulary measures, with decisive evidence and a higher correlation (*r* = 0.57 to 0.68) between EVT, MRT tasks, and EHC visuomotor processing. This was further supported by regression analyses indicating that EVT performance was the strongest unique predictor of multisensory MRTs, EHC, and RAN time. Additionally, visual MRTs were found to predict both receptive and expressive vocabulary. The findings of the study have important implications as accessible school-based assessments of the concurrent development of NVIQ, language, and multisensory processing; and hence as rapid and timely measures of developmental and neurodevelopmental status.

## 1. Introduction

Language is defined as a system of communication to facilitate social interaction and self-expression [1,2] that incorporates symbols, gestures, and sounds, including spoken and written words (vocabulary) or icons. Language development begins in early infancy through toddlerhood with the gradual learning and understanding of single words (receptive vocabulary) and continues with the more rapid acquisition of semantic understanding of words that can express desires or ideas (expressive vocabulary) through childhood into adolescence [3,4,5]. However, what is less well understood is the concurrent development of visually driven intelligence and language acquisition and goal-directed multisensory actions in early school-age children.

Indeed, early developmental research by Kail (1994) proposed that both the linguistic and cognitive processing of children with Specific Language Impairments (SLIs) were characterized by the generalized domain-general deficit ‘slowing’ of reaction times (RTs) on nonverbal tasks (involving visual or/and auditory stimuli) [6]. This hypothesis has since been supported by a number of cognitive studies including a recent meta-analytical review of 46 published studies in children (mean age 8.9 years) that found that individuals with Developmental Language Disorder (DLD) exhibit slower motor RTs in nonverbal tasks, which contributed to observed deficits in language processing, motor skills, and executive functioning [7]. Additionally, a longitudinal study noted that a faster and more accurate performance in *looking-while-listening* tasks in infants aged 15, 18, 21, and 25 months was associated with a faster and more accelerated maturation in expressive vocabulary across the second year of life [8]. Similarly, the early achievement of sensorimotor milestones and early language learning including word acquisition has been reported [9], while multisensory motor reaction times (MRTs) and vocabulary word number at 25 months have been shown to predict later cognitive outcomes, such as generalised intelligence using the Mental Processing Index (MPI) and working memory measures at 8 years of age [10]. Gross motor abilities [11,12] and multisensory attention skills [13] have also been found to predict receptive and expressive vocabulary performance in children aged 1 to 5 years (for a review, see [14]), while automated eye-tracking technology has demonstrated an association between audiovisual asynchrony processing in speech perception tasks and scores on measures of the receptive and expressive language abilities of young children aged 1 to 7 years [15].

The idea of domain-general development has been confirmed by brain imaging by Imada et al. (2006), that reported that the sensory-motor system already begins developing rapidly at around 5–6 months of age and that the neural networks underlying multisensory motor and language information processing in the superior temporal and inferior frontal region [16] in infants and adults are linked to the motor system through multiple connections between the dorsal and ventral prefrontal and premotor cortices [17,18]. However, the relationships between the development of multisensory vision/hearing, mouth/tongue vocalization motor skills, and language processing have seldom been investigated in neurotypical school-age children, though our recent studies [19,20] have identified age, nonverbal intelligence (NVIQ) on the Raven’s Progressive Colored Matrices (RPCM), and visual working memory as the strongest predictors of multisensory MRT tasks.

Thus, the aim of this study was to explore the concurrent development of classical measures of receptive (Peabody Picture Vocabulary Test (PPVT)) and Expressive Vocabulary Test (EVT) performance, as well as the speed of multisensory MRTs processing in terms of age and NVIQ of neurotypical young children (5–10 years). The current study also aimed to use basic motor reaction times to visual, auditory, and audiovisual stimuli and other more complex cognitively associated measures of time (e.g., time taken to complete the motor tracing of shapes in a novel Eye–Hand Coordination (EHC SLURP) task), and visual–verbal speed which was assessed using the Rapid Automatic Naming (RAN) test of familiar objects. Motor and cognitive processing speeds have previously been shown to decrease with age, i.e., performance improves with age [19,20]. The RAN test was included as a quantitative measure of early visual-verbal processing, i.e., total time needed to firstly visually process a familiar expected object and secondly to access the lexical storage system [21,22], rather than as a traditionally predictive measure of reading performance and other language-related tasks in dyslexia [23] and deficits in phonological processing [24].

Building upon these findings [24,25], the current study sought to enhance the understanding of the concurrent development of classical vocabulary measures (e.g., PPVT, EVT, NVIQ, multisensory processing, and EHC SLURP) in three groups of early school-aged children (5–6, 7–8, and 9–10). Specifically, our aims were as follows:

To investigate the apparent developmental changes in vocabulary measures using PPVT and EVT tests and the RAN task. It was hypothesized that children in all groups would demonstrate significant improvements in language and RAN measures with increasing age. In line with the findings of Reinhartsen et al. (2019) [25], we expected the children’s receptive and expressive vocabulary tests would demonstrate age-related developmental trajectories.To explore the relationships between age, NVIQ, MRT measures of multisensory processing EHC, and classical vocabulary measures. Based on both the generalized slowing hypothesis by both Kail (1994) and LeBarton and Iverson (2013) [6,26], we hypothesized that age and higher raw scores on the NVIQ would be highly associated with a more complex expressive vocabulary (as opposed to a more simple receptive vocabulary) and faster MRTs in multisensory tasks and the completion of EHC SLURP items.Lastly, to investigate whether performance in vocabulary tasks predicts MRT measures of multisensory processing and EHC SLURP and whether simple multisensory processes (measured as MRTs, EHC SLURP and RAN) are predictive of developmental vocabulary measures. We hypothesized that a measure of expressive vocabulary (EVT) that requires verbal expression and the integration of visual perceptual and auditory output [5] would contribute more to the rate of multisensory and visuo-motor processing than receptive language. On the basis of our previous study [20], we also expected that visual MRTs would contribute more to vocabulary measures (PPVT and EVT) than auditory MRTs.

## 2. Method

### 2.1. Participants

In this study, a total of 75 participants (59% male) enrolled in foundation/Prep year to Grade 4 were recruited from both Catholic and Public Elementary Schools in metropolitan Melbourne, Australia. The participants were categorized into three age groups: 5–6 years (*n* = 25), 7–8 years (*n* = 26), and 9–10 years (*n* = 24). Ethical approval for the project was obtained from the Human Ethics Committee of La Trobe University (HEC 18139, HEC 16121), Victorian Department of Education and Catholic Education Melbourne. Individual school principals assisted with the distribution of study information and consent forms to parents and guardians. The inclusion criteria were as follows: children between the ages of 5 and 10 years who showed normal or corrected-to-normal vision and hearing, along with adequate color vision, and no clinical diagnosis of neurodevelopmental disorders such as language impairments, autism spectrum disorder (ASD), or intellectual disability, as indicated by a non-verbal IQ (NVIQ) standard score ≥ 85. The children who took part in the research were restricted to those whose parents gave consent by signing the forms stating that “my child is allowed to participate in the study” and also filling out the accompanying questionnaire about their child’s medical history and any possible developmental problems. Children’s verbal consent to the testing was ascertained prior to each testing session. Under the Helsinki Declaration, withdrawal of permission to participate was available to all parents or children at any time.

### 2.2. Screening and Psychometric Tests

#### 2.2.1. Vision and Hearing Screening

Hearing and vision screenings were conducted to ensure that the children had normal hearing and normal-or-corrected to normal vision. First, vision screening involved assessing distance and near visual acuity with Lea Symbols chart [27], while color vision was assessed with Ishihara tests. Second, during the auditory screening, a commercially available portable audiometer (Interacoustic Screening Audiometer, model AS208) manufactured by Interacoustic (A/S, Assens, Denmark) with Peltor (H7A) sound-attenuating headphones with a frequency range of (250–8000 Hz) and 20 dB sound pressure levels for each octave were used to assess a child’s hearing ability. Hearing screening procedures were followed based on the Guidelines for the School Setting of Hearing Screening, Division of Community and Public Health, Missouri Department of Health and Senior Services.

#### 2.2.2. Nonverbal Intelligence (RCPM)

NVIQ of all participants was evaluated using Raven’s Colored Progressive Matrix (RCPM) [28]. In addition to being a relatively quick, well-normed test able to measure nonverbal reasoning abilities in Australian schoolchildren [29], the RCPM is also accepted internationally as demonstrating highly reliable measures of nonverbal intellectual ability in children aged 5–11 [30]. The selection of this test was based on its culture-free items and the fact that cognitive visual manipulation is required rather than language cues. RCPM was administered as an untimed test divided into three sets, each containing 12 problems that progressively increase in complexity and difficulty. For each item, participants were required to choose what they thought was the best of six alternatives available to complete the matrix. Factor analysis indicates that the RCPM measures four distinct intellectual abilities: proficiency in completing simple continuous patterns, proficiency in completing discrete patterns, proficiency in completing simple and complex structures, and proficiency in reasoning by analogy [31,32], which makes the test a very good measure of nonverbal intelligence for problem solving.

### 2.3. Experimental Measures

#### 2.3.1. Multisensory Task

To assess the multisensory processing threshold, we used a target detection task that involved measuring the speed of participants MRTs. The methodology employed for this task was based on prior research [33] and our own earlier studies [19,20]. The task presented three types of stimuli: an auditory stimulus alone (AS; beep), a visual stimulus alone (VS; gray circle), and both audio-visual stimulus (AVS; beep and gray circle presented simultaneously) (see Figure 1). To indicate the stimulus and record responses, children were instructed to press a button as rapidly and accurately as possible on the handheld RESPONSEPixx button box (Model VPX-ACC-3100). VPixx^TM^ software (V 3.20) and RESPONSEPixx hardware (VPixx, Vision Science Solutions, Quebec, Canada) were used to present and control stimuli for the task. To ensure that all participants, particularly those in the youngest age group, understood and could accurately perform the task, a practice trial was conducted for all three types of stimuli (AS, VS, and AVS) prior to testing. Closed headphones were used to present the auditory stimulus (AS) consisting of a 1500 Hz tone with a 5 ms rise and fall time. To ensure conscious attention during the task, the visual target stimulus was displayed as a Gaussian circle presented at various positions on the screen (away from the center). The mean MRTs for each condition of the task, i.e., visual-only, auditory-only, and audiovisual stimuli, were determined as the time interval between the onset of a stimulus and the button press response. Only MRTs scores within the range of 150 ms to 1500 ms were considered when calculating the mean reaction time for each participant. In terms of accuracy, error rates below 50% (i.e., seven out of fifteen errors) were excluded from the analysis. The multisensory task demonstrated high internal reliability with a Cronbach alpha score of 0.93, and the scores for AS, VS, and AVS ranged from 0.87 to 0.9 [19].

#### 2.3.2. Visuomotor Processing Using the SLURP Eye–Hand Coordination (EHC) App

The Lee–Ryan Eye–Hand Coordination Test [34] (SLURP) was used to evaluate fine visually driven motor ability (visuomotor). SLURP was purchased from Apple App Store for $2.10 USD (dated 13 August 2020), and it has been shown that this task is a reliable valid and effective measure of visuomotor integration skills in both children and adults [34,35]. The task requires children to trace five shapes in a particular order after practicing a Castle shape. The Castle item was selected to familiarize children with the procedure and because this relatively difficult item requires many changes in direction while tracing across an iPad screen of 12 inches [35] (see Figure 2). For each child, the time taken to complete five shapes (circle, triangle, square, rabbit, and snail) was extracted and analyzed.

#### 2.3.3. Rapid Automatized Naming (RAN) Task

The RAN was originally developed to measure the speed and accuracy of continuous naming responses [23,24,36]. The RAN tasks [37] used in this study consisted of 36 objects of 6 randomly repeated objects (i.e., boat, star, pencil, chair, fish, and key) (Figure 3). It has been suggested that RAN can be used for both visual–verbal (language domain) [38] and processing speed [21] contributions to reading. Participants were required to verbally name each object from left to right as fast and accurately as possible. To ensure the consistency of naming trials, the task begins with a practice trial consisting of all six familiar objects to ensure that all participants understood the names of objects and instructions of the task performance. The time and errors made in naming all objects were recorded. Only the total time score (i.e., how fast participants can verbally name the objects) was analyzed from this task. The RAN task has been demonstrated to be highly reliable (test–retest r = 0.90) [39].

#### 2.3.4. Receptive Vocabulary Task

Receptive vocabulary ability was measured using the Peabody Picture Vocabulary Test, Fourth Edition (PPVT-4) [40] (Dunn and Dunn, 2007). Children were asked to point to the picture that matched the spoken word from an array of four colored pictures. Responses were scored dichotomously, meaning that they were either correct or incorrect. This test comprises 192 target words in 12-item sets of increasing difficulty. For the starting and ending items, we followed the ceiling and basal set rules to ensure that the examinee only receives sets appropriate to their vocabulary level. The PPVT-4 is an untimed test and has demonstrated highly reliable estimates within the normative samples (a *=* 0.95) [40].

#### 2.3.5. Expressive Vocabulary Task

Expressive vocabulary ability was measured by using the Expressive Vocabulary Test, Second Edition (EVT-2) [41], which was co-normed with the PPVT-. Children were asked to provide a short verbal description or the most appropriate single-word synonym that described the picture. Responses were scored dichotomously (either one or zero). This test continues until five consecutive errors are made or until the entire test is completed. The EVT-2 is an untimed test that has demonstrated highly reliable estimates within the normative samples (a = 0.94) [41].

### 2.4. Procedure

A child was assessed if their guardian returned a signed ethics consent form to the school. Vision and hearing screening was conducted first, followed by adequate practice trials for all experimental tasks. Assessments were conducted individually during school hours in a quiet private room in the presence of at least two researchers. Sessions were limited to 20–30 min in length, with assessments typically conducted over three or four sessions to ensure engagement and reduce fatigue of participants. In cases where children were unable to focus on the tasks, at any time, children were encouraged to take a break or return to class. At the end of each session, each child received a sticker or small stationery item as a thank you.

### 2.5. Data Screening and Analysis

Power analysis. The sample size for the number of participants was estimated by power analysis using the G*Power 3.1 analysis software [42]. According to Cohen (1992), in order to reach moderate effect sizes with α < 0.05 and a power of 0.8 (1-β error probability when conducting one-way ANOVAs, a sample size of 32 participants is recommended for frequentist analyses [43].

Data Cleaning and Outliers. For multisensory MRT tasks, appropriately timed MRT responses were recorded and averaged for each participant, following the exclusion of reaction times below 150 ms or above 1500 ms, as suggested by previous studies [44,45]. The extremely slow RTs indicated participant inattention; however, extremely fast RTs indicated either a false alarm or a response to a previous stimulus [44]. According to these criteria, only 1% of the RT responses were excluded. Two children (in the 5–6 and 7–8 groups) made more than 50% errors in multisensory trials, so their data were excluded. For the RAN task, boxplots identified one outlier whose data were removed. No exclusion was necessary for either the EHC SLURP or vocabulary tasks. According to Victorian school class medians, we divided the participants into three categories based on age (5–6, 7–8, and 9–10 years). All participants were measured for NVIQ in order to ensure they were within the range of normal IQ (see Table 1).

Data Analysis. A Bayesian statistical approach using JASP 0.16.3.0 free software (JASP Team, 2022; http://www.jasp-stats.org/, accessed on 2 February 2023 [46]) was used to analyze all data. Bayesian statistics and analysis were chosen due to their theoretical and practical advantages in the assessment and interpretation of developmental data [47] and in order to facilitate straightforward interpretation [48]. Bayesian statistics are not based on the assumption of normality; thus, they demonstrate important advantages in use with small samples [48]. Such analyses facilitate the assessment of the strength of evidence for each model using a model comparison and selection strategy rather than the null hypothesis testing models associated with frequentist statistics [49,50]. It has also been reported that Bayesian statistics can be used to conduct multiple statistical tests without increasing the risk of Type 1 Errors [51]. Bayes factors (BF_10_) greater than zero were considered evidence in favor of alternative hypotheses. Based on Wetzels and Wagenmakers (2012) [52], BF_10_ values were interpreted as anecdotal evidence if values were between BF_10_ 1 and 3, moderate evidence if values were between BF_10_ 3 and 10, strong evidence if values were between BF_10_ 10 and 30, very strong evidence if values were between BF_10_ 30 and 100, and extreme or/decisive evidence if BF_10_ values were 100 or above.

Data analysis was performed using Bayesian for ANOVA, correlations, and multiple linear regressions.

Firstly, Bayesian one-way ANOVAs were conducted to examine whether there were differences in performance between the three age groups in vocabulary measures (PPVT and EVT) and RAN task. To obtain post hoc comparisons for each Bayesian ANOVA, a default *t*-test with a Cauchy prior was utilized, as suggested by Wagenmakers et al. (2018) [53]. Posterior odds estimates and 95% credible intervals (95%CI) have been reported. In addition, Omega-squared (
$\omega^{2})$
 were calculated to estimate the effect size (ES: 
$\omega^{2}$
 > 0.01 = small; 
$\omega^{2}$
 > 0.06 = moderate; 
$\omega^{2}$
 > 0.14 = large) for differences between groups and to ensure a less biased estimation of variance [54,55,56].

Secondly, Correlations were conducted to investigate the associations between the multisensory processing (AS, VS, and AVS), visuomotor performance, language measures (PPVT and EVT), and RAN task. Bayesian correlations were determined using the default prior (“stretched beta prior width” = 1.0). Pearson correlation coefficients (*r*) and the Bayes Factor (BF_10_) are reported in this study.

Lastly, two directions of Bayesian linear Regressions analyses were also performed to determine (i) the predictive value of language and vocabulary development to MRT measures of multisensory processing (i.e., auditory RT, visual RT, audiovisual RT, and EHC SLURP) and (ii) the predictive value of simple multisensory processes measured as auditory RT, visual RT, audiovisual RT, EHC SLURP, and RAN to language and vocabulary development. In the first regression analysis, we entered the vocabulary tests (PPVT and EVT) and RAN task scores as predictor variables. In the second regression analysis, we entered the simple MRT, EHC SLURP, and RAN as predictor variables. For all models of regression, we presented the Bayes factor (BF) in comparison to the best fitting model, as well as the BF_inclusion_, which indicated that values above 1 suggested that predictors should be included (all values reported are detailed in Goss-Sampson, 2019 [57]).

## 3. Results

### 3.1. Results 1: Age-Group Differences in Receptive and Expressive Vocabulary Tests (PPVT and EVT) and Rapid Automatized Naming (RAN) across Three Age Groups

A series of Bayesian one-way ANOVAs were performed to determine whether there were age-related differences in scores in the Receptive and Expressive Vocabulary Tests (PPVT and EVT) and Rapid Automatized Naming (RAN) tasks for the three age groups. The descriptive statistics for all dependent measures are shown in Table 2. Results of Bayesian one-way ANOVA of vocabulary measures showed decisive evidence for differences across groups, favoring the alternative hypothesis (BF10 = 4.264 × 10^7^, ω2 = 0.48, BF10 = 1.239 × 10^7^, ω2 = 0.49) for PPVT and EVT, respectively. For the PPVT task, post hoc analysis showed that these significant differences were primarily driven by the youngest age group (5–6-year-old group) performing decisively worse, i.e., 5–6 year olds had smaller vocabularies than the children form 7–8 and 9–10 year old groups. The findings also indicate moderate evidence of differences between the 7–8 age group and the 9–10 age group (see Figure 4a and Table 3a). For the EVT task, post hoc comparisons showed that there were very strong to decisive differences between the three age groups (see Figure 4b and Table 3b). For the RAN task, results revealed strong evidence for differences among the groups, supporting the alternative hypothesis (BF10 = 25.777, ω2 = 0.16). Post hoc comparisons indicated that the 9–10 age group exhibited faster performances regarding the RAN test compared to the 5–6 age group. However, there was no evidence indicating differences between the 7–8 age group and either the 5–6 or 9–10 age groups (see Figure 4c and Table 3c), nor was there any evidence of any sex-related differences. Analyses and results of sex-related differences can be found in Appendix A.

### 3.2. Results 2: Relationships among Age, NVIQ, and Vocabulary Tests (PPVT, EVT) and Multisensory MRT Tasks

Bayesian correlations were performed to investigate the strength of evidence for associations between chronological age, NVIQ, vocabulary tasks, and timed measures of multisensory processing (visual, auditory audiovisual, EHC SLURP, and RAN) in this sample of young early school-age children. First, we found evidence of a correlation between chronological age and all of our dependent measures (r = 0.55–0.76), supporting the alternative hypothesis, with a significant negative Pearson’s correlation observed between MRTs in the multisensory task, EHC SLURP, and RAN with age, indicating age-related decreases in the time required to complete visually driven motor tasks. Second, there was *very strong* evidence to suggest that faster MRT tasks were associated with higher NVIQ scores, as well as better performance in PPVT and EVT tasks, with the highest correlation (*r* = 0.70) being between EVT and NVIQ, indicating that a higher NVIQ score is *decisively* associated with greater expressive vocabulary ability. Our results also demonstrated that EVT was *decisively* correlated with all multisensory MRT tasks, EHC, and RAN (*r* = 0.48–0.63), suggesting that better performance in the expressive vocabulary task is correlated with faster MRTs in multisensory tasks such as visual RT, auditory RT, Audiovisual RT, visuomotor processing. *Very strong* evidence of relationships was also found between the receptive vocabulary (PPVT) task and multisensory MRT to only VS, AVS, and EHC SLURP tasks. Lastly, results showed that timing in the RAN task significantly correlated with EHC timing in SLURP (very strong r = 0.45), supporting the hypothesis that better performance in both tasks requires faster visual perception and faster motor responses (Table 4).

Separate Bayesian correlation analyses for each age group (5–6, 7–8 and 9–10 years) were conducted to ascertain whether these associations differ by age group. For the 5–6 years group, the results revealed that there was only anecdotal evidence of associations between NVIQ, EHC SLURP, and EVT task. For the 7–8 years group, anecdotal to moderate evidence of the association was found between NVIQ, multisensory MRTs to VS and AVS, and both PPVT and EVT tasks. For children in the 9–10 age group, there is a notable trend of increasing correlations, as interpreted from anecdotal to very strong evidence, between NVIQ, multisensory MRTs, RAN, and EVT. There was also moderate to very strong evidence between the EVT task and multisensory MRT tasks, EHC and RAN, suggesting that better performance in expressive vocabulary tasks would be associated with faster MRTs of multisensory tasks in older children. In Appendix A, full correlation tables are provided for each age group.

### 3.3. Results 3: Receptive and Expressive Vocabulary Tests and Rapid Automatized Naming (RAN) Predict Multisensory MRT Measures and EHC SLURP and Vice Versa

Bayesian linear regressions were performed next to determine whether receptive and expressive vocabulary tasks and RAN predict multisensory processing (i.e., auditory RT, visual RT, audiovisual RT, and EHC SLURP) (see Table 5) and whether multisensory processes (i.e., auditory RT, visual RT, audiovisual RT, and EHC SLURP) and RAN predict receptive and expressive vocabulary tasks (see Table 6).

Firstly, to determine whether performance in vocabulary tests predicts multisensory MRT tasks, three regression models (PPVT, EVT, and RAN) were used to predict auditory RT, visual RT, audiovisual RT, and total time to complete each item in the visuomotor task (EHC) (see Table 5). In all regression analyses, the EVT scores were the best predictive model. For auditory and audiovisual RT, the odds (BF_M_) in favor of the model containing EVT as a predictor increased by a factor of 5.45 and 10.96, respectively. This model was 1.23 times for auditory RT and 2.94 times more likely than the model with the next highest BF_10_ value. Similarly, for visual RT, EVT increased the likelihood of the model containing it by a factor of (BF_M_ = 15.54), and it was 5.07 times more likely than the model with the next highest BF_10_ value. For the EHC the visuomotor task, the best predictive model was for both EVT+ RAN; the likelihood of the model with both EVT and RAN as predictors increased by (BF_M_ = 3.81), making the model 1.01 times more likely compared to the model with the next highest BF_10_ value. Table 7 provides the posterior summary for Bayes factor inclusion and shows *decisive evidence* for the inclusion of EVT in this model as a predictor of all multisensory MRTs. There is also evidence that EVT (*moderate*) and RAN (*only anecdotal*) should be included as predictors for the visuomotor EHC Slurp task items.

As shown in Table 5e, RAN was considered as a visuo-verbal motor task and thus regressed in PPVT and EVT task performance. Similarly, the best predictive model was composed of the EVT scores. The odds of this model were increased by (BF_M_ = 9.29), making this model 3.38 times more likely than the next highest BF_10_ value, and posterior summary of this model indicated *very strong* evidence for the inclusion of EVT as a predictor. Overall, the expressive vocabulary task (EVT), which is a language measure of visually derived information, was a unique and constant predictor of multisensory MRTs, EHC visuo-motor processing, and visual–verbal–motor RAN task performance.

To determine whether the rate of multisensory processing predicts scores in receptive and expressive vocabulary tasks, five regression models investigating multisensory MRTs to AS, VS and AVS, visuomotor (EHC), and RAN task as predictors of receptive vocabulary (PPVT) and expressive vocabulary (EVT) were examined (Table 6). In the PPVT test, the best predictive model was the time score of the visual RT, as the odds (BF_M_ = 10.92) favored the model including visual RT as a predictor. In the EVT test, the best predictive model was also the scores regarding the time taken to complete the visual RT and EHC SLURP tests; the odds (BF_M_ = 10.71) favored the model including this model as a predictor. Table 8 provides the posterior summary for Bayes factor inclusion. Results showed *moderate evidence* for visual RT to be included as a predictor of both PPVT and EVT, while there was only *anecdotal evidence* for the RAN to be included as a predictor of expressive vocabulary (EVT). Overall, visual RT was found to be a consistent predictor of vocabulary measures. Since we have previously demonstrated that age and NVIQ were the most influential predictors of both multisensory motor tasks and cognitive abilities such as working memory [19,20], and as supported by our correlation analysis (refer to Table 2), we intentionally excluded them from our regression analysis to focus on other variables and obtain more meaningful results. However, detailed regressions involving age, NVIQ, and their prediction of MRT measures of multisensory processing can be found in the Appendix A.

## 4. Discussion

The primary objective of the current study was to investigate the concurrent development of receptive and expressive vocabulary performance, NVIQ, multisensory MRTs processing regarding visual, auditory, audiovisual stimuli, and fine visual motor processing in early school-age children. Collectively, the results of the different age groups (5–6, 7–8, and 9–10 years old) indicated significant and very strong to decisive evidence of improved performance in EVT among the three age groups, while there was moderate evidence of differences in PPVT, confirming different developmental patterns. Our results also indicated that there was evidence for associations between chronological age, NVIQ, vocabulary measures, MRT in multisensory tasks, and visuomotor EHC. Our findings also showed that complex EVT performance, which predominantly requires verbal expression combined with visual perceptual input and auditory output, was decisively correlated with multisensory MRT tasks, EHC, and RAN tasks. This was further supported by our Bayesian regression analyses, as we found that EVT performance (not PPVT) was a unique and constant predictor of multisensory MRTs, visuo-motor processing, and RAN. Such results suggest that increasing the complexity of expressive vocabulary skills involving verbalization using visual inputs and auditory outputs plays a more critical role in multisensory processing rather than simply understanding the meaning of words (i.e., receptive vocabulary skills). In subsequent sections, age-group-related differences in vocabulary measures and the RAN task will be discussed first, followed by the associations between vocabulary measures, NVIQ, and MRTs in multisensory tasks.

### 4.1. Age-Group Differences in Receptive and Expressive Vocabulary Tests and RAN

Consistent with our hypotheses, significant age-group differences were demonstrated in both receptive and expressive vocabulary tests. More specifically, receptive vocabulary performance, as measured by the PPVT test was only different for children aged 5–6 years old but not children aged 7-year-old and above, suggesting that receptive vocabulary development may plateau in the later stages of childhood. In addition, we found very strong to decisive differences between the three age groups for expressive vocabulary measured via the EVT task, which indicates that expressive vocabulary skills continue to develop throughout childhood. This is consistent with previous research in the literature that has shown that the highest rate of oral vocabulary growth, according to the PPVT, test occurs during preschool ages from birth until 5–6 years of age, and this rate declines for each subsequent age period [59,60]. A recent study conducted by Acha et al. (2023) has also demonstrated that the language system, which incorporates a phonological component with storage, monitoring, and sentence-processing abilities, is relatively well-developed in 6 to 7-year-old children [61]. The authors of this study also suggested that receptive vocabulary skills develop earlier than expressive vocabulary skills [62], which is clearly in line with the developmental sequence of these skills (i.e., receptive and expressive skills) [63] in preverbal children. There is also evidence suggesting that children in the first years of life tend to depend more on their ability to understand linguistic information, but at a later age, the maturation of expressive skills becomes more important for a comprehensive understanding of pictures and symbols [64]. These results are further supported by a recent systematic review conducted by Dobinson and Dockrell (2021) [65] that examined the importance of using universal strategies (i.e., structured vocabulary programs and approaches involving speech and language therapists) in order to improve expressive rather than receptive language skills during the early school years [65], whereby a child’s expressive language is closely associated with improved literacy and education outcomes in primary school [66,67,68].

Visual (first process) and verbal (second process) processing, as measured by the RAN task, also showed significant differences (i.e., strong evidence) between the 5–6 and 9–10 age groups, indicating that the older children were significantly faster at naming objects compared to the younger children. This finding is consistent with those of Alghamdi et al. (2021) [21] and Peters et al. (2020) [69], who found statistically significant differences in children aged 5–8 years for RAN performance. One possible interpretation of these results could be based on previous research [69,70,71] that has revealed the significance of attention and higher cognitive processes in eye movement-driven temporal processing during Rapid Automatized Naming (RAN) tasks for successful object recognition. Consequently, it is possible that the faster naming of stimuli observed in older children in the current study is indicative of their superior rapid sequential modality shift processing skills [72].

### 4.2. Age, NVIQ, and Their Relationship with Multisensory MRT Tasks and Vocabulary Tasks

In line with our current hypotheses and our previous work [24] we found significant correlations between age, NVIQ, and decreased MRTs in the multisensory tasks involving audiovisual, visuo-motor processing, and visual–verbal (RAN) processing. Furthermore, children with higher NVIQ and working memory scores showed faster performance in multisensory MRT tasks [20]. In the current study, NVIQ was also significantly correlated with vocabulary tasks (PPVT and EVT), which is in line with the body of evidence that indicates that receptive and expressive vocabulary tests are highly correlated with performance on the measures of NVIQ in both adults and children [32,73,74].

More interestingly, our hypothesis regarding the association between vocabulary tests and novel multisensory MRTs and EHC SLURP was supported by our Bayesian analyses, as we found evidence of an association between the PPVT, EVT tests, and multisensory RT to visual, auditory, audiovisual, and visuomotor tasks. Once again, the EVT task showed a more decisive and higher correlation with multisensory MRT tasks than the PPVT task. These findings lend support to Kail’s ‘generalized slowing hypothesis’ [6], which suggests that the differences in processing speed, as measured by motor reaction time (RT) tasks, between children with Specific Language Impairments and those without, are not specific to the task itself but instead reflect a more general cognitive processing component [6]. Indeed, Haapala et al. (2014) have also noted that better linguistic skills, such as reading fluency and comprehension, are associated with better motor performance [75]. This association is thought to be related to the overlap of brain networks, such as the inferior frontal gyrus and left superior temporal gyrus, which are involved in both visuo-motor and verbalization processing [76]. The activation and maturation of these shared brain regions [77] presumably facilitates the development of both motor and linguistic skills and, equally likely, the slower neurodevelopment of such areas presumably accounts for the executive dysfunction observed in individuals of all ages diagnosed with neurodevelopmental disorders such as dyslexia [78].

Our findings also fit well with previous evidence that indicates that faster processing speeds in nonverbal motor tasks are associated with language development in early infancy [79,80] and in school-age children (ages 8–14 years) for both expressive and receptive language skills [81,82]. One possibility is that faster looking time, i.e., time required for an infant to fixate an object and then to ability to rapidly shift eye gaze and hence attention, reflects the earlier development of automatization regarding both the motor control of eye-driven attention and extraction of salience from a simple range of tasks [80]. Gaze patterns involve the integration of selective attention and the perceptual and receptive processing of instructions and prescribed targets to support the integration of auditory and visual information and motor processing, thereby enhancing language development [26,79]. Presumably, the automatization of visually or verbally driven motor actions is closely related to modality shift effect research [72,83] that refers to the time and neural resources required to shift between different sensory modalities (e.g., auditory and visual) while performing cognitive tasks, which should facilitate word learning in the earliest stages of receptive language development.

### 4.3. Predictive Ability of Receptive and Expressive Vocabulary Scores for Multisensory MRT Tasks and Vice Versa

In our Bayesian regression analyses, we found that the EVT test consistently predicted the rate of multisensory MRTs and EHC, supporting the hypothesis that expressive rather than receptive vocabulary tests would contribute to multisensory processing tasks. This finding is consistent with a study by Peter et al. (2019), who found that the processing speed of spoken word recognition in toddlers (assessed by the *looking-while-listening paradigm*) predicted children’s expressive vocabulary and utterance complexity during early development [80]. Indeed, our findings add to the large body of studies that support the notion that individuals with faster MRTs (i.e., faster switching from a distractor to a target image) have larger expressive vocabularies than those with slower MRTs in both children [8,10,61,79,84] and adults [85,86]. This may be due to some developmental differences in cognitive abilities, such as NVIQ, attentional resources, perceptual processing, which would contribute to the associations between the greater speed and accuracy of MRTs and growth in expressive language [84].

In addition, the findings reported in the current study demonstrate that the measure of speed of MRTs to visual stimuli has consistently predicted scores on vocabulary tasks, as measured by PPVT and EVT. This finding suggests that individuals with faster visual motor processing abilities are more likely to perform better on vocabulary tasks [87], which is in line with previous research in infants [8,10]. In addition, Yu et al. suggested that visually driven sustained attention with longer eye fixation on objects by infants leads to more successful word learning [88,89]. Such observations are in line with previous electrophysiological and psychophysical research showing that the fast-conducting Magnocellularly driven attention of the dorsal brain networks is associated with better NVIQ scores and reading [69], motor coordination, and cognitive abilities [71,90,91].

## 5. Limitations

A particular strength of this study is the use of a variety of visuomotor tasks (i.e., simple multisensory motor reaction time task, EHC, and visual–verbal motor reaction time (RAN) task) to measure multisensory motor processing in children and the utilization of Bayesian probability statistics in accordance with recent analytical recommendations to assess the strength of evidence for the alternative hypothesis [92,93]. On the other hand, an important limitation of the current study is that the selective classical measures of receptive and expressive vocabulary tests, such as the Peabody Picture Vocabulary Test (PPVT) and the Expressive Vocabulary Test (EVT), use pictorially derived contexts to assess verbal language understanding (i.e., receptive language) and expression rather than more extensive measurements of language manipulation skills and verbal comprehension abilities, e.g., grammar and sentence processing abilities. Thus, future studies may benefit from including other aspects of language abilities tasks, as well as a parent report of language, such as the Alberta Language and Development Questionnaire [94], to provide a more comprehensive understanding of children’s language skills.

A further limitation of the present study was the absence of an independent assessment of non-motor components of multisensory auditory and visual threshold detection times. Thus, future research should aim to include non-motor reaction times for both visual and auditory recognition tasks in addition to using other robust measures of oculomotor function, such as eye movement or/and flicker fusion thresholds [69], to assess the recovery time of visual conduction pathways between the eye and cortex. Finally, it is also important to acknowledge that although the sample size of the current study was not large, the use of Bayesian analysis, which does not assume normality, was a statistical advantage [48].

## 6. Conclusions and Future Directions

To our knowledge, our study is the first to investigate whether the time required to respond to multisensory information and the completion of EHC SLURP items predicts classical measures of vocabulary development, including receptive and expressive performance in early elementary school-age children. Overall, our findings using Bayesian analyses provide evidence for age-group differences in expressive vocabulary performance, as measured by EVT test, while PPVT performance was only significantly different for children aged between 5 and 6 years old but not children aged 7 and above, highlighting different developmental trajectories of types of language acquisition during early childhood. Furthermore, our results show evidence for associations between chronological age, NVIQ, MRT in multisensory tasks, EHC, and vocabulary measures of both PPVT and EVT, with there being a decisive correlation between EVT and all MRT tasks. Our most important finding indicates that EVT performance (but not PPVT) is a unique and constant predictor of multisensory MRTs, visuo-motor processing for EHC tasks, and visual–verbal processing using the RAN. Our results are unique in the sense that they provide preliminary evidence that increasing the complexity of expressive vocabulary skills, which involves combining visual and auditory processing with verbal expression rather than simply understanding the meaning of words (i.e., receptive vocabulary skills), contributes more significantly to the rate of multisensory processing and acquisition of fine motor skills. However, future research using more precise measures of specific aspects of automatization of expressive language, such as syntactic complexity and lexical skills rather than just extent of lexical vocabulary, could provide insight into the relationships between expressive vocabulary skills and multisensory processing. Results from the current study also have educational implications in terms of providing an easy-to-administer, accessible, school-based system to assess the concurrent development of NVIQ and language and multisensory processing. The aforementioned factors can all be better assessed through the use of PPVT, EVT, and SLURP Eye–Hand Coordination test. All tests are readily attainable and are rigorous time-sensitive measures that can be used to identify developmental and neurodevelopmental conditions.

## Figures and Tables

**Figure 1 brainsci-13-00965-f001:**
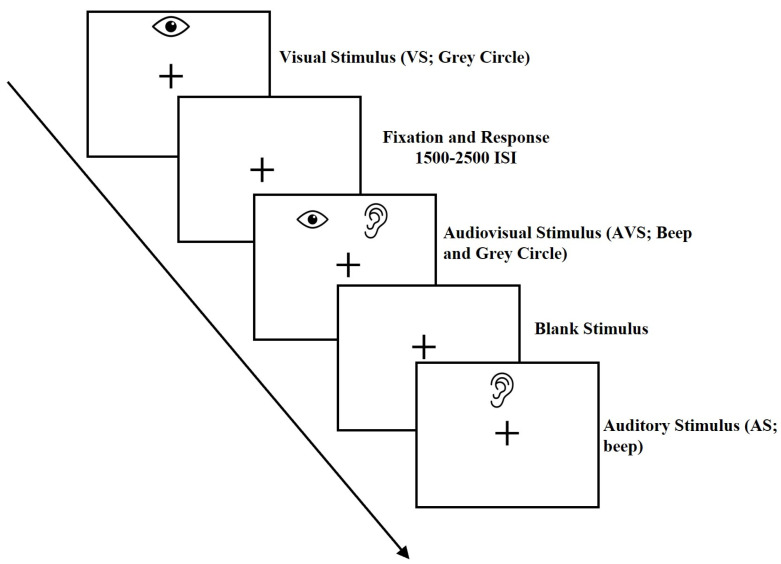
An example of three types of stimuli (AS, VS, and AVS) used in multisensory tasks.

**Figure 2 brainsci-13-00965-f002:**
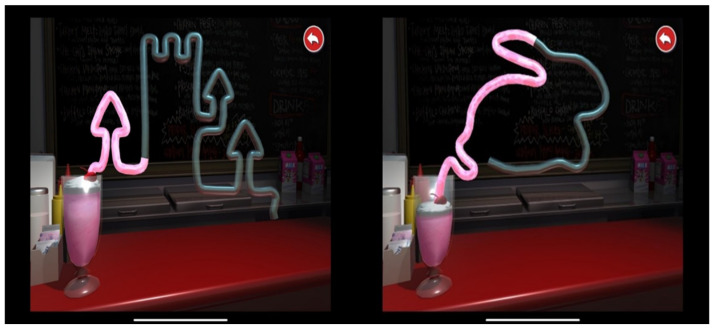
Two examples of Eye–Hand Coordination Test (SLURP).

**Figure 3 brainsci-13-00965-f003:**
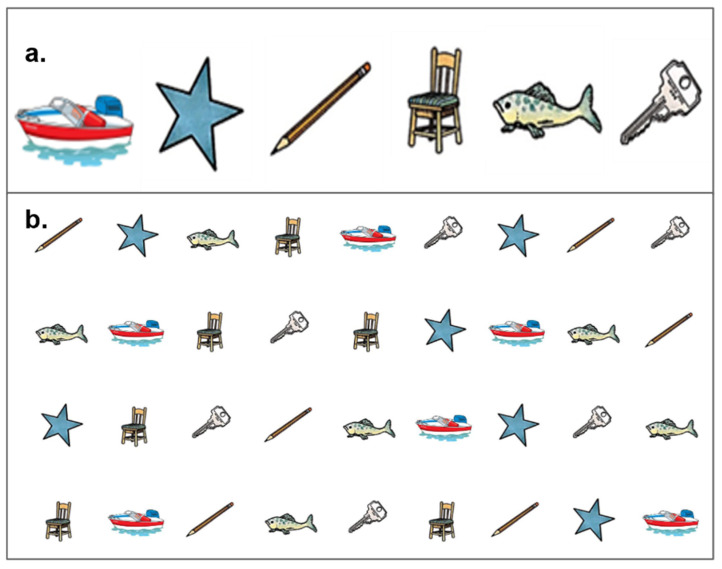
Example of Rapid Automatic Naming (RAN) practice trial (**a**); timed trial (**b**).

**Figure 4 brainsci-13-00965-f004:**
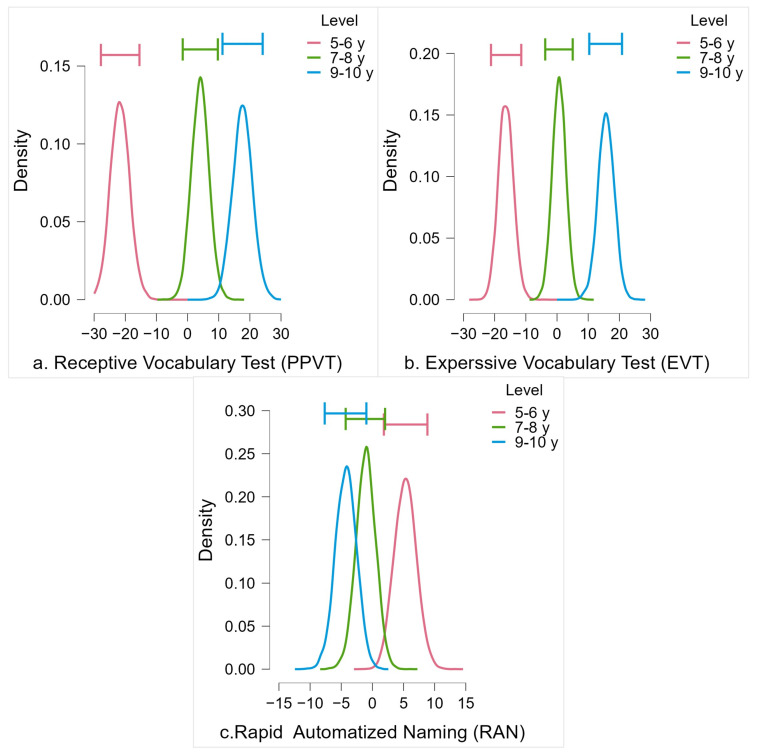
The model-averaged posterior distribution (horizontal bars show the 95% credible intervals around the median) for (**a**) Receptive Vocabulary Test (PPVT), (**b**) Expressive Vocabulary Test (EVT), and (**c**) Rapid Automatized Naming (RAN).

**Table 1 brainsci-13-00965-t001:** The descriptive statistics present the mean age (SD), raw scores, and standard score for NVIQ measure in each age group.

N		Age Range			NVIQ (RS)			NVIQ (SS)		
		Min.	Max.	M (SD)	Min.	Max.	M (SD)	Min.	Max.	M (SD)
5–6 years	24	5.00	6.90	6.00 (0.58)	11.00	29.00	17.91 (5.07)	86.00	130.00	102.87
										−11.64
7–8 years	25	7.00	8.79	7.93 (0.48)	20.00	34.00	26.48 (3.78)	89.00	128.00	109.84
										−10.18
9–10 years	24	9.00	10.99	9.94 (0.66)	26.00	34.00	30.04 (2.56)	89.00	121.00	106.73
										−9.05
Total	73									

Note: NVIQ= Non-verbal IQ assessed using Raven’s Colored Progressive Matrices (RCPM), and scores range from 0 to 36; RS = Raw Score of RCPM; SS = Standard Score of RCPM.

**Table 2 brainsci-13-00965-t002:** Descriptive statistics for PPVT, EVT, and RAN by age groups.

95% Credible Interval					
Measure	Age	M	SD	Lower	Upper
PPVT	5–6 years 7–8 years 9–10 years	116.304 143.115 157.611	18.247 13.765 18.983	108.414 137.556 148.171	124.195 148.675 167.051
EVT	5–6 years 7–8 years 9–10 years	83.524 101.269 117.267	11.075 13.367 14.023	78.482 95.870 109.501	88.565 106.668 125.032
RAN (ms)	5–6 years 7–8 years 9–10 years	47.117 39.569 35.848	8.798 9.650 10.312	43.000 35.396 31.275	51.235 43.742 40.420

Note: PPVT = Overall scores of the Peabody Picture Vocabulary Test; EVT = Overall scores of Expressive Vocabulary Test (EVT); RAN = Rapid Automatized Naming (RAN) response time.

**Table 3 brainsci-13-00965-t003:** Post hoc comparisons.

		Prior Odds	Posterior Odds	BF_10, U_	Error %
PPVT					
5–6 years	7–8 years	0.587	15,426.617	26,262.495	1.148 × 10^−10^
	9–10 years	0.587	298,595.504	508,333.280	4.549 × 10^−11^
7–8 years	9–10 years	0.587	4.728	8.049	1.190 × 10^−6^
b. EVT					
5–6 years	7–8 years	0.587	722.460	1229.926	9.769 × 10^−9^
	9–10 years	0.587	2.028 × 10^6^	3.452e+6	1.839 × 10^−9^
7–8 years	9–10 years	0.587	21.318	36.292	6.217 × 10^−7^
c. RAN					
5–6 years	7–8 years	0.587	1.136	1.935	0.008
	9–10 years	0.587	32.742	55.741	1.185 × 10^−7^
7–8 years	9–10 years	0.587	0.439	0.747	0.007

Note: The posterior odds have been corrected for multiple testing by fixing the prior probability to 0.5 so that the null hypothesis holds across all comparisons [58]. Individual comparisons are based on the default *t*-test with a Cauchy (0, r = 1/sqrt (2)) prior. The “U” in the Bayes factor denotes that it is uncorrected.

**Table 4 brainsci-13-00965-t004:** Bayesian Pearson Correlations for Total Sample.

Variable		Age	RCPM	AS	VS	AVS	EHC	PPVT	EVT	RAN
1. Age	Pearson’s r	—								
	BF₁₀	—								
2. RCPM	Pearson’s r	0.764 ***	—							
	BF₁₀	2.081 × 10^12^	—							
3. AS	Pearson’s r	−0.552 ***	−0.410 **	—						
	BF₁₀	42,147.607	83.288	—						
4. VS	Pearson’s r	−0.664 ***	−0.568 ***	0.774 ***	—					
	BF₁₀	7.570 × 10^7^	100,894.359	7.069 × 10^12^	—					
5. AVS	Pearson’s r	−0.686 ***	−0.559 ***	0.816 ***	0.872 ***	—				
	BF₁₀	4.976 × 10^8^	62,846.539	4.156e × 10^5^	4.116 × 10^20^	—				
6. EHC	Pearson’s r	−0.688 ***	−0.560 ***	0.516 ***	0.572 ***	0.605 ***	—			
	BF₁₀	1.936 × 10^7^	9011.275	1287.939	16,180.995	89,593.183	—			
7. PPVT	Pearson’s r	0.715 ***	0.633 ***	−0.351	−0.506 ***	−0.421 **	−0.450 **	—		
	BF₁₀	4.985 × 10^8^	938,481.963	8.672	1319.799	60.777	61.115	—		
8. EVT	Pearson’s r	0.751 ***	0.679 ***	−0.578 ***	−0.639 ***	−0.621 ***	−0.564 ***	0.823 ***	—	
	BF₁₀	2.169 × 10^9^	5.527 × 10^6^	14,531.478	404,008.271	142,794.927	2096.183	3.054 × 10^13^	—	
9. RAN	Pearson’s r	−0.505 ***	−0.377 *	0.362 *	0.327	0.344	0.451 **	−0.359	−0.487 ***	—
	BF₁₀	1230.234	16.746	11.192	4.921	7.305	76.554	7.291	194.699	—

Note: Age = age in numbers; RCPM = nonverbal IQ of Raven; AS = auditory MTR stimuli; VS = visual MRT stimuli; AVS = audiovisual MRT stimuli; EHC SLURP = visuomotor tasks; PPVT = Peabody picture vocabulary test; EVT = expressive vocabulary test; RAN = rapid automatized task response time. * BF₁₀ > 10, ** BF₁₀ > 30, *** BF₁₀ > 100.

**Table 5 brainsci-13-00965-t005:** Multiple Bayesian regressions for PPVT, EVT, and RAN predicting Multisensory MRTs to Auditory, Visual, and Audiovisual stimuli and EHC SLURP.

Model Predictors	P (M)	P (M|Data)	BF_M_	BF_10_	R^2^
Auditory RT					
EVT	0.125	0.438	5.456	1.000	0.308
EVT + PPVT	0.125	0.355	3.860	0.811	0.343
EVT + RAN	0.125	0.105	0.820	0.239	0.309
EVT + PPVT + RAN	0.125	0.099	0.768	0.226	0.343
PPVT	0.125	0.001	0.008	0.002	0.120
PPVT + RAN	0.125	9.590 × 10^−4^	0.007	0.002	0.158
RAN	0.125	5.888 × 10^−4^	0.004	0.001	0.097
Null model	0.125	1.993 × 10^−4^	0.001	4.550 × 10^−4^	0.000
b. Visual RT					
EVT	0.125	0.689	15.540	1.000	0.406
EVT + RAN	0.125	0.136	1.101	0.197	0.406
EVT + PPVT	0.125	0.135	1.088	0.195	0.406
EVT + PPVT + RAN	0.125	0.034	0.244	0.049	0.407
PPVT	0.125	0.005	0.032	0.007	0.277
PPVT + RAN	0.125	0.002	0.012	0.003	0.292
RAN	0.125	1.736 × 10^−5^	1.215 × 10^−4^	2.518 × 10^−5^	0.095
Null model	0.125	6.249 × 10^−6^	4.374 × 10^−5^	9.063 × 10^−6^	0.000
c. Audiovisual RT					
EVT	0.125	0.610	10.960	1.000	0.366
EVT + PPVT	0.125	0.207	1.828	0.339	0.379
EVT + RAN	0.125	0.127	1.018	0.208	0.366
EVT + PPVT + RAN	0.125	0.054	0.398	0.088	0.379
PPVT	0.125	0.001	0.007	0.002	0.185
PPVT + RAN	0.125	6.295 × 10^−4^	0.004	0.001	0.210
RAN	0.125	8.289 × 10^−5^	5.803 × 10^−4^	1.358 × 10^−4^	0.095
Null model	0.125	2.976 × 10^−5^	2.083 × 10^−4^	4.877 × 10^−5^	0.000
d. EHC SLURP					
EVT + RAN	0.125	0.353	3.811	1.000	0.334
EVT	0.125	0.347	3.722	0.985	0.289
EVT + PPVT + RAN	0.125	0.104	0.816	0.296	0.334
EVT + PPVT	0.125	0.090	0.691	0.255	0.290
PPVT + RAN	0.125	0.069	0.523	0.197	0.281
RAN	0.125	0.023	0.164	0.065	0.196
PPVT	0.125	0.013	0.091	0.036	0.174
Null model	0.125	8.332 × 10^−4^	0.006	0.002	0.000
e. RAN					
EVT	0.250	0.756	9.291	1.000	0.258
EVT + PPVT	0.250	0.223	0.859	0.295	0.263
PPVT	0.250	0.019	0.059	0.026	0.140
Null model	0.250	0.002	0.006	0.003	0.000

Note: EVT = expressive vocabulary test; PPVT = Peabody picture vocabulary test; RAN = rapid automatized task response time; EHC SLURP = visual motor processing.

**Table 6 brainsci-13-00965-t006:** Multiple Bayesian regressions for Multisensory MRTs to Auditory, Visual, and Audiovisual stimuli, EHC SLURP, and RAN predicting PPVT and EVT.

Model Predictors	P (M)	P (M|DATA)	BF_M_	BF_10_	R^2^
PPVT					
VS	0.031	0.260	10.920	1.000	0.290
VS + EHC SLURP	0.031	0.112	3.925	0.431	0.308
VS + RAN	0.031	0.095	3.262	0.366	0.302
AS + VS	0.031	0.077	2.588	0.296	0.296
VS + AVS	0.031	0.067	2.240	0.259	0.291
AS + VS + EHC SLURP	0.031	0.043	1.400	0.166	0.315
VS + RAN + EHC SLURP	0.031	0.041	1.316	0.156	0.313
VS + AVS + EHC SLURP	0.031	0.040	1.276	0.152	0.312
AS + VS + RAN	0.031	0.038	1.235	0.147	0.311
VS + AVS + RAN	0.031	0.032	1.026	0.123	0.306
b. EVT					
VS + EHC SLURP	0.031	0.257	10.717	1.000	0.545
VS + RAN + EHC SLURP	0.031	0.155	5.670	0.602	0.566
VS + RAN	0.031	0.104	3.614	0.406	0.526
VS + AVS + EHC SLURP	0.031	0.060	1.980	0.234	0.547
AS + VS + EHC SLURP	0.031	0.058	1.921	0.227	0.547
VS	0.031	0.057	1.874	0.222	0.474
VS + AVS + RAN + EHC SLURP	0.031	0.041	1.309	0.158	0.569
AS + VS + RAN + EHC SLURP	0.031	0.039	1.243	0.150	0.568
VS + AVS + RAN	0.031	0.034	1.075	0.130	0.535
AS + VS + RAN	0.031	0.027	0.854	0.104	0.530

Note: VS = visual RT; AS = auditory RT; AVS = audiovisual RT; EHC SLURP = visual motor processing; RAN = rapid automatized task response time; EVT = expressive vocabulary test; PPVT = Peabody picture vocabulary test; RAN = rapid automatized task response time.

**Table 7 brainsci-13-00965-t007:** Posterior summaries of regression coefficients for PPVT, EVT, and RAN predicting Multisensory MRTs to Auditory, Visual, and Audiovisual stimuli and EHC SLURP.

Coefficient	P (incl)	P (incl|Data)	BF_inclusion_	Mean	SD	95% Credible Interval	
						Lower	Upper
Auditory RT							
Intercept	1.000	1.000	1.000	869.537	16.728	835.300	904.376
EVT	0.500	0.997	351.833	−5.118	1.596	−8.782	−2.523
PPVT	0.500	0.456	0.839	0.875	1.266	−0.203	3.891
RAN	0.500	0.205	0.258	0.082	0.795	−1.921	2.120
b. Visual RT							
Intercept	1.000	1.000	1.000	904.763	14.019	878.190	931.600
EVT	0.500	0.994	156.342	−4.464	0.966	−6.298	−2.558
PPVT	0.500	0.175	0.212	−0.034	0.483	−1.455	0.878
RAN	0.500	0.171	0.207	−0.039	0.610	−1.751	1.070
c. Audiovisual RT							
Intercept	1.000	1.000	1.000	824.211	13.577	798.533	852.953
EVT	0.500	0.998	551.686	−4.227	1.020	−6.502	−2.271
PPVT	0.500	0.263	0.356	0.254	0.674	−0.299	2.608
RAN	0.500	0.182	0.222	−0.004	0.597	−1.197	1.835
d. EHC SLURP							
Intercept	1.000	1.000	1.000	67.373	2.266	62.898	71.910
EVT	0.500	0.894	8.431	−0.404	0.206	−0.711	0.000
PPVT	0.500	0.277	0.382	−0.016	0.101	−0.337	0.168
RAN	0.500	0.549	1.219	0.230	0.274	−0.059	0.789
e. RAN							
Intercept	1.000	1.000	1.000	41.040	1.268	38.434	43.496
EVT	0.500	0.979	45.733	−0.287	0.092	−0.488	−0.116
PPVT	0.500	0.242	0.319	0.009	0.054	−0.107	0.171

Note: EVT = expressive vocabulary test; PPVT = Peabody picture vocabulary test; RAN = rapid automatized task response time; EHC SLURP = visual motor processing.

**Table 8 brainsci-13-00965-t008:** Posterior summaries of regression coefficients for Multisensory MRTs to Auditory, Visual, and Audiovisual stimuli, EHC SLURP, and RAN predicting PPVT and EVT.

Coefficient	P (incl)	P (incl|Data)	BF_inclusion_	Mean	SD	95% Credible Interval	
						Lower	Upper
PPVT							
Intercept	1.000	1.000	1.000	138.113	2.686	132.533	143.765
AS	0.500	0.269	0.368	0.004	0.017	−0.021	0.061
VS	0.500	0.903	9.294	−0.066	0.033	−0.121	0.000
AVS	0.500	0.289	0.406	−0.001	0.027	−0.068	0.063
RAN	0.500	0.297	0.422	−0.065	0.176	−0.588	0.187
EHC SLURP	0.500	0.343	0.522	−0.056	0.118	−0.386	0.077
b. EVT							
Intercept	1.000	1.000	1.000	100.373	1.799	96.802	103.841
AS	0.500	0.204	0.257	−0.002	0.011	−0.033	0.016
VS	0.500	0.904	9.374	−0.062	0.028	−0.099	0.000
AVS	0.500	0.279	0.388	−0.009	0.024	−0.085	0.017
RAN	0.500	0.471	0.889	−0.154	0.211	−0.655	0.006
EHC SLURP	0.500	0.692	2.250	−0.167	0.143	−0.415	0.000

Note: VS = visual RT; AS = auditory RT; AVS = audiovisual RT; EHC SLURP = visual motor processing; RAN = rapid automatized task; EVT = expressive vocabulary test; PPVT Peabody picture vocabulary test; RAN = rapid automatized task response time.

## Data Availability

All data are available upon request.

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
