# Peer review of "Visual Motor Reaction Times Predict Receptive and Expressive Language Development in Early School-Age Children"

_brainsci, 2023, doi:10.3390/brainsci13060965_

Round 1

Reviewer 1 Report

Comments and Suggestions for Authors

The paper “Visual Motor Reaction Times Predict Receptive and Expressive Language Development in Early School-Age Children” investigates the relationship between visuo-motor and linguistic performance. To this aim, the authors analyzed data from verbal, non-verbal, and visuo-motor testing in 75 children. The results showed the correlation between vocabulary production and motor reaction time, supported by additional regressions and prediction. The study sounds timely and worth. The results are straightforward. The manuscript offers a general overview of the findings. There might be room for streamline the introduction and better insert the study in a broader framework.

1) The introduction seems to be at the limit of length, risking to overload the reader, which might feel lost within too much information. It is suggested to streamline and shrink the introduction in order to easier guide the reader through the most important concepts up to the logic flow that brings from the need to the rationale of the study.

2) It has been recently suggested that better linguistic skills (e.g. reading) are associated with more precise motor performance (e.g. haptic exploration) (Longcamp et al 2005 Acta Psych), and that both are bound to the activity of sets of brain regions that are not limited to the visual cortex (Brozzoli et al 2019 Front Psychol), and whose correct maturation is essential from early age on (Ionta 2021 Front Human Neur). A similar neurally-based framework might strengthen the discussion of the present study’s results, in that it would provide the background to support that the brain activity associated with motor performance and linguistic skills is at least partially overlapping (Halje et al 2015 Neuropsychologia). Providing a similar, more mechanistic, interpretation of how the maturation of the motor system could explain the benefits in linguistic skills found in the present study, could be beneficial to better understand how the present study sits in a broader framework.

3) From a clinical standpoint, it might be worth noting the association between motor and linguistic impairments in neurological, psychological, and emotional disorders. Respectively, (1) the prefrontal regions typically linked to motor skills, also have been associated with dysfunctional reading, such as dyslexia, especially in teenagers and adults (Farah et al 2021, Frontiers in Psychology), (2) cognitive impairment can affect the excitability of the motor cortex associated specifically with linguistic tasks, and (3) depression affects both verbal fluency and motor performance. This suggests that brain-based mechanisms sustain the motor-linguistic association found in the present study, and that a failure in one of these domains is reflected in the other one. In other words, providing a more brain-based interpretation of the relationship between motor and linguistic skills could be beneficial to better understand how the present study sits in a broader framework.

Author Response

To Assistant Editor, and the Reviewer,

Thank you for the opportunity to resubmit a revised draft of our manuscript for publication in the Sensory and Motor Neuroscience section of the Journal of Brain Sciences. We sincerely appreciate the valuable assessment provided by the Reviewers. Each comment has been thoroughly considered and addressed individually to the best of our understanding, as well as in the track-changed revisions to the manuscript.

Reviewer #1

The paper “Visual Motor Reaction Times Predict Receptive and Expressive Language Development in Early School-Age Children” investigates the relationship between visuo-motor and linguistic performance. To this aim, the authors analyzed data from verbal, non-verbal, and visuo-motor testing in 75 children. The results showed the correlation between vocabulary production and motor reaction time, supported by additional regressions and prediction. The study sounds timely and worth. The results are straightforward. The manuscript offers a general overview of the findings. There might be room for streamline the introduction and better insert the study in a broader framework.

(1) The introduction seems to be at the limit of length, risking to overload the reader, which might feel lost within too much information. It is suggested to streamline and shrink the introduction in order to easier guide the reader through the most important concepts up to the logic flow that brings from the need to the rationale of the study.

Author Response: 

We sincerely thank the reviewer for their valuable comments! 

1) Thank you for pointing this out. We have now modified our introduction and reduced the text -especially on page 2 paragraph 3 - to improve clarity for the readers. All changes are highlighted in the manuscript.

2) It has been recently suggested that better linguistic skills (e.g. reading) are associated with more precise motor performance (e.g. haptic exploration) (Longcamp et al 2005 Acta Psych), and that both are bound to the activity of sets of brain regions that are not limited to the visual cortex (Brozzoli et al 2019 Front Psychol), and whose correct maturation is essential from early age on (Ionta 2021 Front Human Neur). A similar neurally-based framework might strengthen the discussion of the present study’s results, in that it would provide the background to support that the brain activity associated with motor performance and linguistic skills is at least partially overlapping (Halje et al 2015 Neuropsychologia). Providing a similar, more mechanistic, interpretation of how the maturation of the motor system could explain the benefits in linguistic skills found in the present study, could be beneficial to better understand how the present study sits in a broader framework.

3) From a clinical standpoint, it might be worth noting the association between motor and linguistic impairments in neurological, psychological, and emotional disorders. Respectively, (1) the prefrontal regions typically linked to motor skills, also have been associated with dysfunctional reading, such as dyslexia, especially in teenagers and adults (Farah et al 2021, Frontiers in Psychology), (2) cognitive impairment can affect the excitability of the motor cortex associated specifically with linguistic tasks, and depression affects both verbal fluency and motor performance. This suggests that brain-based mechanisms sustain the motor-linguistic association found in the present study, and that a failure in one of these domains is reflected in the other one. In other words, providing a more brain-based interpretation of the relationship between motor and linguistic skills could be beneficial to better understand how the present study sits in a broader framework.

Author Response: 

2) + 3) We thank the reviewer for these valuable suggestions and refs. We have now added this information as suggested in the discussion section on page 19 line 8 to read

“These findings lend support to the ‘generalized slowing hypothesis’ of Kail [6], which suggests that the differences in processing speed, as measured by motor reaction time (RT) tasks, between children with Specific Language Impairment and those without it, are not specific to the task itself but instead reflect a more general cognitive processing component [6]. Indeed, Haapala et al. (2014) have also noted that better linguistic skills such as reading fluency and comprehension are associated with better motor performance (Haapala et al., 2014). This association is thought to be related to the overlap of brain networks such as inferior frontal gyrus and left superior temporal gyrus, that are involved in both visuo-motor and verbalization processing (Halje et al., 2015). The activation and maturation of these shared brain regions (Ionta, 2021) presumably facilitates the development of both motor and linguistic skills and equally likely slower neurodevelopment of such areas presumably accounts for the retarded executive function observed in individuals of all ages diagnosed with neurodevelopmental disorders such as dyslexia (Farah et al., 2021)”.

We have now added more references as the reviewer suggested (see below): 

Farah, R., Ionta, S., & Horowitz-Kraus, T. (2021). Neuro-Behavioral Correlates of Executive Dysfunctions in Dyslexia Over Development From Childhood to Adulthood. Frontiers in Psychology, 12. doi:10.3389/fpsyg.2021.708863

Haapala, E. A., Poikkeus, A.-M., Tompuri, T., Kukkonen-Harjula, K., Leppänen, P. H. T., Lindi, V., & Lakka, T. A. (2014). Associations of motor and cardiovascular performance with academic skills in children. Medicine and science in sports and exercise, 46(5), 1016-1024. doi:10.1249/mss.0000000000000186

Halje, P., Seeck, M., Blanke, O., & Ionta, S. (2015). Inferior frontal oscillations reveal visuo-motor matching for actions and speech: evidence from human intracranial recordings. Neuropsychologia, 79, 206-214.

Ionta, S. (2021). Visual Neuropsychology in Development: Anatomo-Functional Brain Mechanisms of Action/Perception Binding in Health and Disease [Review]. Frontiers in Human Neuroscience, 15.

Reviewer 2 Report

Comments and Suggestions for Authors

- The IRB numbers “HEC 18139, HEC 16121” were already associated with a previous study. Could the authors explain if the project was written the multiple study publication?

- It is advised to explain who performed the different scales and if permission was requested for their use. The complete description of who performed and the permission should also be described for the software used.

- Was permission requested to use the figures provided in the manuscript? It is advised to write in the figure legend due to copyright issues.

- The power of the study should be based on a previous study assessing a similar idea. The study provided “Cohen et al.” is a general description.

- Why was JASP used instead of other, more frequently used statistical software?

- How was the distribution of the variables?

- How were selected the variables for the regression?

- What were the models used in the regression? None? Backward?

- How did the authors control confounding variables?

- Was the difference between sex analyzed?

- In the table with regressions, it is advised only to include the statistically significant results.

Others

- The authors should provide references in the “Instruction for authors” style.

- Revise auto-citations. It is advised to maintain at most 10% of auto citations. There are 13 references with the third author (CSG).

- How this manuscript differs from others already published by the author? Alhamdan, A.A.; Murphy, M.J.; Pickering, H.E.; Crewther, S.G. The Contribution of Visual and Auditory Working Memory and Non-Verbal IQ to Motor Multisensory Processing in Elementary School Children. Brain Sci. 2023, 13, 270. https://doi.org/10.3390/brainsci13020270

Author Response

To Assistant Editor, and the reviewer,

Thank you for the opportunity to resubmit a revised draft of our manuscript for publication in the Sensory and Motor Neuroscience section of the Journal of Brain Sciences. We sincerely appreciate the valuable assessment provided by the Reviewers. Each comment has been thoroughly considered and addressed individually to the best of our understanding, as well as in the track-changed revisions to the manuscript.

1-The IRB numbers “HEC 18139, HEC 16121” were already associated with a previous study. Could the authors explain if the project was written the multiple study publication?

Author Response: 1-We thank the reviewer for raising this question. We would like to clarify that our previous study also utilized the same ethics approvals "HEC 18139" and "HEC 16121", for research into multisensory processing and language in school age children and which extend over 3 years minimum duration.  In Australia it is common for ethics applications to encompass a sequence of multiple experimental protocols under one project, which have to be evaluated and approved before schools and parents can be approached and prior to recruitment or testing and data collection for multiple associated studies.  Thus, there was no impact on participant testing and each study approved under these IRB/HEC numbers conducted unique research procedures and analyses, focusing on different aspects of the research question.

2-It is advised to explain who performed the different scales and if permission was requested for their use. The complete description of who performed and the permission should also be described for the software used.

Author Response: 2-The information re software used for development of tests is included in the manuscript and all testing was done under the auspices of the Senior Author Prof Sheila Crewther who is a qualified neuropsychologist and neuro-optometrist.

We have also added more information in the Funding section on page 21 to read

“This project was primarily funded by the La Trobe University, School of Psychology and Public Health, Department of Psychology, Counselling and Therapy. The VPixxTM equipment and RESPONSEPixx (VPixx) were funded to Prof. SGC through ARCDP171029. The audiometer used, an Interacoustic Screening Audiometer of portable model AS208 and Peltor H7A sound attenuating headphones, is on loan from Prof. Carl Parsons and the Fildes Foundation.”

3-Was permission requested to use the figures provided in the manuscript? It is advised to write in the figure legend due to copyright issues.

Author Response: 3- We thank the reviewer for this comment. We have now made a new Figure for multisensory task. Thus, all figures now in our manuscript were created specifically for this research by the authors. Please see page 5 Figure 1.

4-The power of the study should be based on a previous study assessing a similar idea. The study provided “Cohen et al.” is a general description.

Author Response: 4- We appreciate the Reviewer’s comment regarding the power of our study and note in text ,page 7 line 4, that we did use G*Power 3.1 analysis software, following the guidelines outlined by Faul et al. (2007) as commonly used in social and behavioral research (Faul, Erdfelder, Lang, & Buchner, 2007).  Cohen’s definitions have been referred to in reference to the specification or the effect sizes i.e., regarding commonly used small, medium, and large effects for the interest of readers more used to frequentist statistics.  Furthermore, such information demonstrates to readers that our design and analysis was sufficiently powered with respect to frequentist methodology. Bayesian statistical analyses do not require power analyses to predict the prior distribution due to the nature of the analyses, in line with this, and published recommendations, we utilised a weakly informative prior (Cauchy’s prior of 0.7) in order to fully assess the study effects with minimal bias (Marsman & Wagenmakers, 2017).

5-Why was JASP used instead of other, more frequently used statistical software?

Author Response: 5- We have chosen JASP software over other commonly used statistical software packages because of several factors including its focus design for use with Bayesian analyses.

First, JASP was specifically designed for Bayesian analyses and is a graphical user interface (GUI) developed using widely tested R-scripts to generate output for interpretation and communication of study outcomes in line with published ‘Bayesian Analysis and Reporting Guidelines (BARG)'. The software is continually evolved to incorporate many features of Bayesian hypothesis testing and Bayesian parameter estimation (Love et al., 2019; Wagenmakers et al., 2018). Many other statistical programs (e.g. SPSS) are only now incorporating Bayesian analysis protocols, and are yet to perform such analyses as comprehensively and generate output as effectively as JASP.

Second, JASP is free and open-source software. As such using this platform supports the development of more inclusive and open science initiatives that do not limit access to data analysis and science communication by expensive paywall barriers.

Lastly, as noted by Marsman and Wagenmakers (2017); Wagenmakers et al. (2018), the JASP GUI is designed to be user-friendly for those familiar with SPSS, and it is built using programming languages such as C++, HTML, and JavaScript. The inferential engine is also based on R (R Development Core Team 2009). Specifically, for the Bayesian analyses, JASP makes extensive use of the Bayes Factor package developed by Morey and Rouder (2015) and Overstall and King (2014), which increases the robustness and reliability of the Bayesian analyses using JASP.

For additional information, please see these two articles

Wagenmakers, EJ., Love, J., Marsman, M. et al. Bayesian inference for psychology. Part II: Example applications with JASP. Psychon Bull Rev 25, 58–76 (2018). https://doi.org/10.3758/s13423-017-1323-7

Marsman, M., & Wagenmakers, E.-J. (2017). Bayesian benefits with JASP. European Journal of Developmental Psychology, 14(5), 545-555. Doi:10.1080/17405629.2016.1259614 

6-How was the distribution of the variables?

Author Response: 6- We thank you for the comment. However as our study, has utilized Bayesian analysis as our primary statistical approach, it is important to note that Bayesian analysis does not rely on assumptions about the specific distribution of variables, as do traditional frequentist methods. Instead, Bayesian analysis focuses on priors and posterior distribution based on observed data which has been reported on page 10 Table (3).

7-How were the variables selected for the regression?

Author Response: 7- Thank you for your comment. Prior to collecting any data or our analyses, we conducted a thorough review of the existing literature related to our research topic and identified variables that have been consistently reported as influential factors or predictors in similar studies with infants and school-age children (Fernald, Perfors, & Marchman, 2006; Marchman & Fernald, 2008; Peter et al., 2019). Adding to this, our previous findings (Alhamdan, Murphy, Pickering, & Crewther, 2023) highlighted that visual rather than auditory processing is the most important cognitive driver associated with simple multisensory MRTs and that further investigation of the relationship of this concept to language was warranted.

Furthermore, prior to conducting the regression analysis, we performed correlation analyses to examine the associations between our variables which showed significant correlations between multisensory motor and vocabulary tasks, this supported our theoretical considerations and justified the inclusion of these variables in the regression analyses.

8- What were the models used in the regression? None? Backward?

Author Response: 8- Thank you for your comment. However, it is important to note that Bayesian linear regression analysis does not typically use the backward or forward regression analysis techniques commonly used in classical (frequentist) linear regression. Bayesian linear regression is built on a different model of probability utilizing probability distributions rather than point estimates.  In this sense the Bayesian regression allows simultaneous testing of the experimental and null hypotheses regarding whether each variable is accounting for the outcome variable or not. This is reflected in the BFM values (favouring contribution for the predictor/predictor combinations) provided in the regression tables (Tables 5 and 7), and the BFinclusion values provided in Tables 6 and 8 accompanying each regression.

In addition, Bayesian linear regression was chosen in our study compared to classical (frequentist) linear regression as it provides several advantages over classical regression techniques, such as “Priors and Posterior Distribution”, and “Model Comparison and Selection” that offers a framework for comparing and selecting between different models based on their posterior probabilities.

9- How did the authors control confounding variables?

Author Response: 9- Thank you very much for this comment. We are unsure which confounding variables (e.g., gender, sex or IQ) he/she is meaning; however, we have now clarified and added more information on page 3 to read

“The inclusion criteria were as follows: children between the ages of 5 and 10 years who showed normal or corrected-to-normal vision and hearing, along with adequate colour vision, and no clinical diagnosis of neurodevelopmental disorders such as language impairments, autism spectrum disorder (ASD), or intellectual disability, as indicated by a non-verbal IQ (NVIQ) standard score ≥85.”

10- Was the difference between sex analyzed?

Author Response: 10- We appreciate the reviewer's comment while noting that although our study did not specifically analyse the differences between sexes, we have rerun the analysis based on gender and found no evidence of sex differences in performance on our tasks. We have now added and reported these results in the supplementary material.

11-In the table with regressions, it is advised only to include the statistically significant results.

Author Response: 11- Thank you for your comment. As Bayesian regression analyses consider probability of evidence for and against the experimental versus the null hypothesis, the results presented allow the reader to determine the value of each predictor and/or model in contributing to the effect of interest, rather than a probability of the data given the hypothesis as with frequentist statistics. It therefore allows the reader to make a judgement regarding the meaningfulness of any given predictor/model based on the Bayes factor given our data.  This is in line with the BARG recommendations noted above.

12-The authors should provide references in the “Instruction for authors” style.

Author Response: 12- Thank you for your comments. We have now checked our references. However, if there is anything specific you would like us to address, please let us know.

13-Revise auto-citations. It is advised to maintain at most 10% of auto citations. There are 13 references with the third author (CSG).

Author Response: 13- The leader of this laboratory group apologizes for having been a researcher in this field for a long time. Where she has self-referenced, it is in regard to the design and development and employment of relevant testing methodologies and variable-defining psychophysical and neuroimaging studies in collaboration with multiple authors. For instance, the references in this manuscript associated with Prof. CSG were shared with over 20 other authors and scientists. Therefore, if there are 13 references with the same authors, it will be considered self-citation.

14-How this manuscript differs from others already published by the author? Alhamdan, A.A.; Murphy, M.J.; Pickering, H.E.; Crewther, S.G. The Contribution of Visual and Auditory Working Memory and Non-Verbal IQ to Motor Multisensory Processing in Elementary School Children. Brain Sci. 2023, 13, 270. https://doi.org/10.3390/brainsci13020270

Author Response: 14. The titles and abstracts of the last paper and this manuscript give evidence to the differences. Our previous study (Alhamdan, Murphy, Pickering, & Crewther, 2023) investigated the influence of working memory (visual and auditory) and nonverbal IQ on multisensory processing. In contrast, the current manuscript explores different cognitive abilities, particularly receptive and expressive language processing, and examines how these contribute to multisensory processing.

References:

Alhamdan, A. A., Murphy, M. J., Pickering, H. E., & Crewther, S. G. (2023). The Contribution of Visual and Auditory Working Memory and Non-Verbal IQ to Motor Multisensory Processing in Elementary School Children. Brain sciences, 13(2), 270. Retrieved from https://www.mdpi.com/2076-3425/13/2/270

Faul, F., Erdfelder, E., Lang, A.-G., & Buchner, A. (2007). G* Power 3: A flexible statistical power analysis program for the social, behavioral, and biomedical sciences. Behavior Research Methods, 39(2), 175-191. doi:https://doi.org/10.3758/BF03193146

Fernald, A., Perfors, A., & Marchman, V. A. (2006). Picking up speed in understanding: Speech processing efficiency and vocabulary growth across the 2nd year. Dev Psychol, 42(1), 98-116. doi:10.1037/0012-1649.42.1.98

Love, J., Selker, R., Marsman, M., Jamil, T., Dropmann, D., Verhagen, J., . . . Wagenmakers, E.-J. (2019). JASP: Graphical Statistical Software for Common Statistical Designs. Journal of Statistical Software, 88(2), 1 - 17. doi:10.18637/jss.v088.i02

Marchman, V. A., & Fernald, A. (2008). Speed of word recognition and vocabulary knowledge in infancy predict cognitive and language outcomes in later childhood. Developmental science, 11(3), F9-F16. doi:https://doi.org/10.1111/j.1467-7687.2008.00671.x

Marsman, M., & Wagenmakers, E.-J. (2017). Bayesian benefits with JASP. European Journal of Developmental Psychology, 14(5), 545-555. doi:10.1080/17405629.2016.1259614

Peter, M. S., Durrant, S., Jessop, A., Bidgood, A., Pine, J. M., & Rowland, C. F. (2019). Does speed of processing or vocabulary size predict later language growth in toddlers? Cognitive Psychology, 115, 101238. doi:https://doi.org/10.1016/j.cogpsych.2019.101238

Team, R. D. C. (2009). A language and environment for statistical computing. http://www. R-project. org.

Wagenmakers, E.-J., Love, J., Marsman, M., Jamil, T., Ly, A., Verhagen, J., . . . Morey, R. D. (2018). Bayesian inference for psychology. Part II: Example applications with JASP. Psychonomic Bulletin & Review, 25(1), 58-76. doi:10.3758/s13423-017-1323-7

Round 2

Reviewer 1 Report

Comments and Suggestions for Authors

accept